# Quality 4.0 and Cognitive Engineering Applied to Quality Management Systems: A Framework

**Adriana Ventura Carvalho**  **and Tânia Miranda Lima \***

C-MAST—Center for Mechanical and Aerospace Science and Technologies, University of Beira Interior, Calçada Fonte do Lameiro, 6200-358 Covilhã, Portugal
\* Correspondence: tmlima@ubi.pt

**Abstract:** In order to create high-quality products, quality engineering must be integrated across the entire product development process. To accomplish the ultimate goal, innovative approaches are required, and a Quality Management System-QMS is imperative to standardize all processes. All business areas depend on people and processes, but quality is especially dependent on them. A QMS can benefit from the application of Quality 4.0—Q4.0 and Cognitive Engineering—CE aspects to reduce the workload and cognitive capacity required from QMS specialists, using these technologies to tackle long-standing quality concerns and to re-optimize to deliver creative solutions. The decision to implement a QMS based on Q4.0 technologies is difficult to take due to the challenge that is to automatize dispersed activities. The purpose of this paper is to develop a framework that aids in the application of a Q4.0 QMS. The relationship between quality management practices and Industry 4.0 technologies that improve quality are deeply studied and connected with CE practices to develop an advanced framework, that makes it easier to overview all the dispersed activities within the manufacturing environment gathered as one, and simplify the application of new technologies to the QMS activities. The proposed framework was developed as result of this study.

**Keywords:** Quality 4.0; Quality Management System; cognitive engineering; Industry 4.0



## 1. Introduction

The growing level of global competition requires that companies find additional ways to remain competitive, and one of the best instruments for businesses to boost competitiveness is a Quality Management System (QMS) [1]. The process of developing a QMS entails, among others, the creation of the organization's quality policy, quality guidelines, recorded procedural documentation for important business operations, internal standards, and the implementation of analytical and statistical control tools [2]. Implementing a QMS, and certifying it thereafter, are voluntary processes backed by the organization's motives, goals, and policies.

In the Industry 4.0 paradigm, globalization has driven businesses to operate in a way that is increasingly complex than it was in the past [3]. This Fourth Industrial Revolution is based on cutting-edge engineering and manufacturing techniques, massive digitization, big data analytics, advanced robotics, adaptive automation, additive and precision manufacturing (e.g., 3D printing), modeling and simulation, artificial intelligence, and nano-engineering of materials [4]. This revolution presents challenges and opportunities.

The recent significant increase in machine learning implementation can be attributed to factors such as easier access to big data sets, increasing computing power and the cognitive load required from workers [5]. It's important to determine which Cognitive Engineering (CE) techniques and strategies are suitable for a QMS context.

A QMS can benefit from the application of Quality 4.0 (Q4.0) and Cognitive engineering aspects to reduce the workload and cognitive capacity required from QMS specialists, using these technologies to tackle long-standing quality concerns and to re-optimize to deliver creative solutions.

The current manuscript is organized as follows: In Section 2 the methodology used is described. In Section 3 the authors present the conceptual framework of the relationship between quality management practices and Industry 4.0 technologies that improve quality and their connection with cognitive engineering practices. To conclude the paper, a discussion of the results and the principal conclusions of the study are made known, followed by the study limitations and future research.

### 1.1. Industry 4.0 and Quality 4.0

Manufacturing may reduce human error, reduce time to market, and hasten the rate at which industrial processes may adapt to new information by using digital technologies. Industry 4.0 describes the real-time synchronization of operational and information technology as well as the interdependence of processes and analytics [6].

Industry 4.0 changes the ways in which businesses produce, improve, and distribute their goods. Manufacturers are incorporating cutting-edge technologies, such as cloud computing and analytics, the IoT, Artificial Intelligence (AI), and Machine Learning (ML), into their manufacturing facilities and operations. In order to create a full and interconnected corporate environment, Industry 4.0, combines physical production and operations with intelligent digital technologies [7]. To achieve this convergence, the manufacturing process must be reconfigured with an architecture that can take in massive amounts of real-time data and allow a better control of the entire environment [8].

Industry 4.0 (I4.0) brings about changes that are significant from an IT standpoint as well as from an organizational one. The ability to decide why and how data should be applied should thus be possessed by Quality experts because the processes, managed by these professionals, must choose how and when to use the information, rather than the opposite way around [9]. Process lifetimes are shortened by technological growth, giving little time to comprehend and address engineering issues from a practical perspective [5].

The term "Quality 4.0" describes the digitization of TQM and how it has affected the quality of procedures and technology, but also how it impacts people. It expands on established quality tools and takes into account connectivity, intelligence, and automation in order to improve performance and make prompt data-driven decisions in an end-to-end scenario that involves all parties and offers visibility and transparency [10]. It might also be described as the use of technology from the fourth industrial revolution to improve quality [11].

Quality 4.0 emphasizes the important details, lowers the cost of subpar quality, and tracks quality outcomes. Using analytics, several organizations have begun to increase the resolution of their data collecting. Instead of focusing on component inspection as the primary quality duty, and besides establishing a Quality Management System (QMS) to themselves, these companies examine the quality and operating practices of their suppliers to prevent issues with quality later on. They understand their suppliers' internal processes and enhance component efficiency in close collaboration with their suppliers' to meet their quality needs [12].

For Quality Management Systems and practitioners to be successful and contribute to the overall advantages of the organization, they must be a seamless part of the environment. Large volumes of data from numerous sources are made available to quality professionals in real-time and concurrently, and this data can be used to enable rapid, tactical decision-making [13]. Each new quality philosophy has altered the approach to problem-solving in order to deal with the industrial systems' ever-increasing complexity [5]. In the fourth industrial revolution era, a brand-new paradigm called Quality 4.0 has recently emerged to address this. The quest of performance excellence during these potentially disruptive times of digital change led to the rise of this paradigm [14].

Quality 4.0 is built on a new paradigm that helps decision makers to make smart decisions based on empirical learning, empirical knowledge discovery, and real-time data generation, collection, and analysis. It makes use of industrial big data, the industrial internet of things, and artificial intelligence, which are I4.0 tools, to address a completely

new set of complex problems [14]. The survival of industry now depends on Quality 4.0. Any process, from the development of products to front-office business activities, must stand out via consistency. Quality 4.0 is more often than not the idea of consistency as a strategy of containment rather than a weapon of prevention in this contemporary period of accelerating change [12]. Quality 4.0 is exciting, and quality manufacturing will adopt at least some of the improvements in the next years. To ensure that quality is given top priority in manufacturing, this quality revolution depends on an unified quality strategy, it attempts to promote quality accountability across the entire supply chain and consumer base [15].

In reality, Quality 4.0 is not about technology, but rather about the people who use it and the methods they employ to get the most out of it [16].

### 1.2. Cognitive Engineering

When deciding how to act, cognition implies the capacity to comprehend the underlying nature of things, including the present, the past, and a forecast of the near future [17]. Analysis, design, and assessment of complex systems involving humans and technology are the focus of the multidisciplinary field of cognitive engineering. A running computer simulation program that implements a broad theory of human cognition based on a variety of human experimental data is known as a cognitive architecture [18].

The purpose of cognitive engineering is to comprehend the problems, demonstrate how to make better decisions when they arise, and outline the trade-offs that must be made when, as is frequently the case, a development in one area results in deficiencies in another [19]. Related to the term Cognitive Systems Engineering, to ensure that cognitive work is effective and reliable, it employs systematic approaches of cognitive analysis and design [20]. Bias influences human decision-making most often [21]. By integrating system functions with the cognitive processes they need to enable, the goal is to increase and extend human ability to know, perceive, decide, plan, act, and collaborate. The system's decision-makers are the people. That demand must come first, then the technology. The goal is to improve workers' ability to think through the design of tools, procedures, or training [20].

We now make decisions more quickly, have continual access to information and data, and distribute our work across individuals and continents thanks to technology. As a result of decreased physical effort and repetitious mental activity, the cognitive component of job is less obvious and tractable [6]. In ways that, until recently, only the human brain could comprehend, cognitive technologies are now capable of identifying significance in this data. In the modern manufacturing environment, where competitiveness and cost sensitivity demand greater levels of agility, reactivity, and creativity from producers, this level of expertise will be seen as vital [22].

The introduction of the new Industry 4.0 paradigm and the quick technological advancements that came with it transformed the role of the employees, who had to switch from simple systems to extremely complex, automated systems. With this change in the role of human operators, it is now more important than ever for them to be able to: comprehend how complex systems, that carry out tasks under their direction, work; quickly acquire and integrate pertinent information; and monitor and correct system flaws [18].

A highly qualified workforce capable of mastering such complex work settings is required for smart factories. Data literacy, abstract thinking, and problem solving are necessary competencies. The concept of a smart factory also foresaw an important role for cognitive support systems that give information for aiding in human decision making. The availability of enormous volumes of information in real-time and need to be matched by appropriate abilities. The capacities to act, think, remember, and infer are referred to as being cognitive functions [23].

An output of knowledge work is the decision making. Decisions and actions are likely to have hidden consequences in complex work contexts, with the real impact only becoming apparent when many system components come together. The context and the interaction of various factors play a role in determining the most effective and value-adding decision rule. Effective approaches to enhance interactions and cognitive efficiency

have been shown through research in the disciplines of human computer interaction and cognitive engineering. Useful models from cognitive engineering can be integrated to create a novel manufacturing framework, or to complement existing ones [24] such as the Quality Management System (QMS).

*1.3. Quality Management System*

The policies, procedures, and controls required for an organization to produce and provide high-quality goods or services to clients, and hence raise customer satisfaction, are documented by a quality management system (QMS) [25].

Growth in globalization supports quality management systems. Only businesses that can match their consumers' expectations will be able to grow as a result of the rising variety and quantity of products, reduced number of essential machines and devices, shorter production times, and shorter product life cycles. Besides testing the quality of products, services and processes to improve them, organizations should also manage systemically if they want to remain competitive [26].

The ability to manage effectively within an organization is made possible by conforming to the requirements of the ISO standards, such as ISO 9001 and ISO 13485, which are based on the PDCA (Plan-Do-Check-Act) cycle. These standards also specify the requirements that must be met in order to guarantee the quality of the processes. Without a QMS in place while executing the organizations processes, it is frequently impossible to operate as a trustworthy partner on the market, especially when dealing with industries whose functioning may pose an high risk, as is, for example the medical devices field (ISO 13485) [27,28].

Risk, which is the impact of uncertainty on the accomplishment of quality objectives, can be identified by a departure from the expected outcome. However, in the same processes, the risk can be established in relation to potential events, their effects, or their combination. As a result, the organization can conduct a qualitative and/or quantitative assessment of risks in the QMS and decide on the best course of action for management [29].

Regardless of size or industry, organizations must apply the risk-oriented approach in the QMS; however, the complexity of the procedures used depends on the maturity of the organization's QMS and the variables in its external environment. The development of the risk-based approach application in the QMS, depends on the significance of these organizational features [30]. The center of quality management operations is the QMS, in recent years the concept of an Electronic QMS (EQMS) started to emerge. It offers a scalable solution to automate workflows, connect quality processes, enhance data integrity, provide centralized analytics, guarantee compliance, and promote collaboration within a single app. Quality affects every step of the value chain and how it is run, making it a hub [16]. This EQMS already falls into the concept of Quality 4.0 and can be explored to fully embrace the available tools.

Better supplier oversight, easier resource management, assurance of customer requirements compliance, and product quality control are all made possible by properly defined procedures and instructions, which further lowers production costs [31].

Although the market has made some strides toward EQMS adoption, many companies remain gravely behind. Organizations put off implementing quality technology in part because their fundamental processes are fragmented; it is difficult to implement technology to automate operations that are fragmented [16].

During this research, the relationship between quality management practices and Industry 4.0 technologies that improve quality were deeply studied and connected with cognitive engineering practices to develop an advanced framework that makes it easier to overview all the dispersed activities within the manufacturing environment gathered as one, and simplify the application of new technologies to the QMS activities.

### 1.4. Scientific Contributions

A variety of techniques and tactics are provided by Quality 4.0 to make monitoring practices simpler. Studying the multiple QMS practices that can be enhanced by cognitive engineering and the relationship between those practices and the Industry 4.0 technologies is necessary. Industries want to automate digital processes and harmonize and connect those automated processes to other systems and activities in order to fully benefit from a QMS. The time required for high-value personnel and management to implement and concentrate on improvement and creativity is reduced when the system is improved. A QMS focused on the outcomes and that provides support throughout the entire PDCA cycle, will encourage behaviors and actions that help the organization produce high-quality results. In order to compete for quality and keep their competitive edge, they must also employ an organizational strategy centered on continual growth.

Several new abilities for quality professionals are essential, and preparation is the key such as having various skills at various levels is appropriate. Using the I4.0QMS, organizations can share more efficiently the regulatory and quality methods across departments and workplaces.

The authors identified as a gap in the literature the lack of contributions relating Quality 4.0 with Quality Management Systems activities, the gap tightens if we add into the equation the cognitive aspects of human workers, that play a crucial role in this new digital era, where it still remains as the most valuable asset. The absence of relevant research in the area also provided insight into the uniqueness and significance of this investigation.

Integrating QMS practices supported by CE techniques with I4.0 technologies and tools helps automate compliance-based tasks and data processing, being these the main research goals of this study. The focus of this study is to develop a framework that can help companies in better assessing their enforcement procedures and identify growth potential.

## 2. Materials and Methods

### 2.1. Research Methodology

The approach introduced in this paper is intended to illustrate how business corporations could achieve a better and more updated Quality Management System (QMS) through the implementation of Industry 4.0/Quality 4.0 tools aligned with the quality management practices and a better application of cognitive associated tasks reducing the load using Cognitive Engineering techniques. The authors introduce a framework that can help corporations in the task of implementing a QMS.

The status review of the I4.0, Q4.0 and Cognitive Engineering topics were performed to find out and merge the existing research knowledge on the concepts related to the QMS tools and technologies.

For a better understanding, the essential information about Quality 4.0, Industry 4.0, Cognitive Engineering and Quality Management System is provided. The goals of the research are made clear by the figures. The development of an I4.0QMS is the main topic of this study. This study will assist further research in better understanding the real world implementation of the framework developed.

A mixed inductive–deductive approach supports this research methodology. An example of the deductive approach is the detailed analysis of the industry 4.0 tools that have a connection with QMS practices (I4.0QMS) and the application of Cognitive Engineering (CE) techniques. In the deductive approach, the dimensions related to the connection between these three major subjects were known. The purpose was to determine how the identified CE techniques have an influence on I4.0 tools and link how these influenced tools will connect with the most common QMS practices in terms of identifying business needs and supporting them throughout the quality process cycle, therefore enhancing planning, implementation, validation and taking action to achieve outstanding and enduring results. ScienceDirect database was used for literature search using the following keywords string "Quality Management System, Cognitive Engineering, Quality 4.0 and Industry 4.0" refining the results by year from 2019 to 2023 and by subject area "Engineering". The articles

that fall outside the scope of this article were also rejected. The number of articles that were included within this research also enhances the novelty and need for this study, since the majority of the articles within the keywords defined fell out of the scope.

The inductive approach involves Industry 4.0, Quality 4.0 and Cognitive Engineering knowledge and related developments. Since the topic of research has a strong practical nature, in addition to papers published in peer-reviewed academic journals, the review also included "gray" literature (e.g., conference proceedings, magazines, and books). The flowchart is represented in Figure 1.

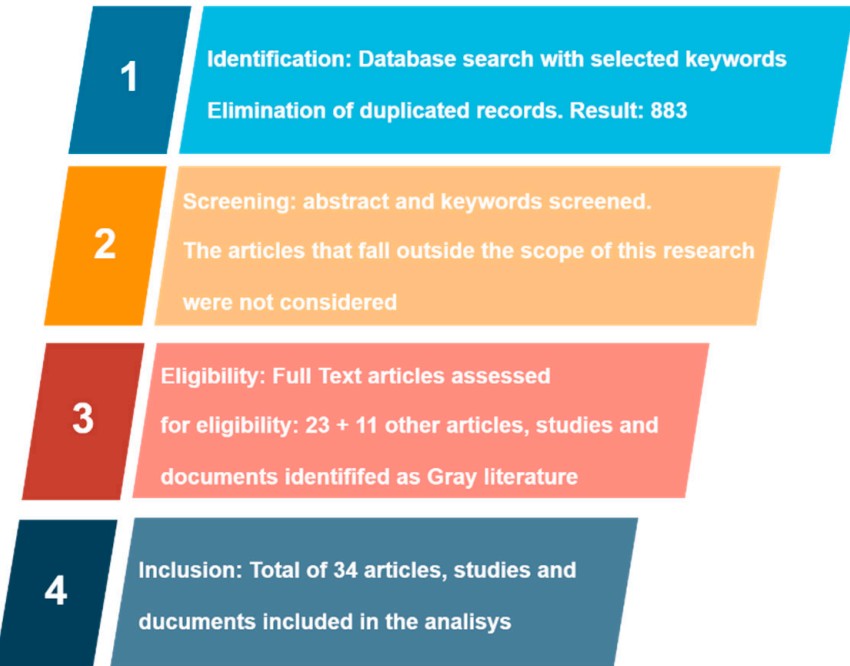

**Figure 1.** Literature Review Flowchart.

Performing this research, the actors acknowledge a lack of recent cognitive engineering research on this field. Using ScienceDirect database and using the keywords "Cognitive Engineering" and refining the results by year from 2019 to 2023 and by subject area "Engineering", only 102 results are available. Using the exclusion criteria of the articles not really mentioning "cognitive engineering" itself or being applicable to fields outside the scope of this article, a very few recent published papers were included in this study. For these reasons, the literature referenced on this article related to this subject are not as recent as the authors aimed for. However, since the most recent research on this field also discusses these articles, the authors deemed them as relevant.

*2.2. Research Objectives*

Implementing a QMS involves careful planning, a solid plan of action, and a thorough comprehension of the elements of the QMS concept that will work best for each specific organization. Organizations can get started by introducing their decision-makers to the following concepts:

1. Target areas where QMS Practices will benefit from cognitive engineering techniques the most.
2. Focus on the core tools of Industry/Quality 4.0 most relevant to the QMS Practices.
3. Align Industry/Quality 4.0 objectives with QMS/Business Goals enhanced with the cognitive engineering basis.

These three concepts constitute the research objectives of this study.

## 3. Results

### 3.1. The Link between Industry 4.0 and QMS Practices

It can be difficult for organizations to make the transition to Quality 4.0. After all, it is a hard departure from conventional methods of putting QMS practices into effect, and the addition of new technology requirements does not make it any simpler.

In the authors previous research this process was simplified [13]. Providing an overview between QMS practices and Industry 4.0 tools and demonstrating a way to link QMS present practices with Industry 4.0 tools.

Anywhere that data is generated inside the organization, it must have some sort of system that collects, saves, and analyses data in real-time, since Quality 4.0's goal is to sustain continuous quality with real-time data [11]. However, what are the I4.0 tools that can be implemented to match QMS practices demands?

The following diagram was performed to better overview the linkage between QMS practices and Industry 4.0 tools (Figure 2), and was developed to answer to the question above. For organizations to achieve a Quality 4.0 ready QMS this diagram will broaden their thinking and help in making associations from what processes they already have implemented and what tools they can start implementing to achieve the ultimate goal—an I4.0QMS. Using the AI I4.0 Tool, for example, it is possible to comply with the Management Commitment, Employee Involvement and Information and Analysis QMS practices, and by doing so a I4.0QMS can be achieved. The same thinking applies to the other identified tools.

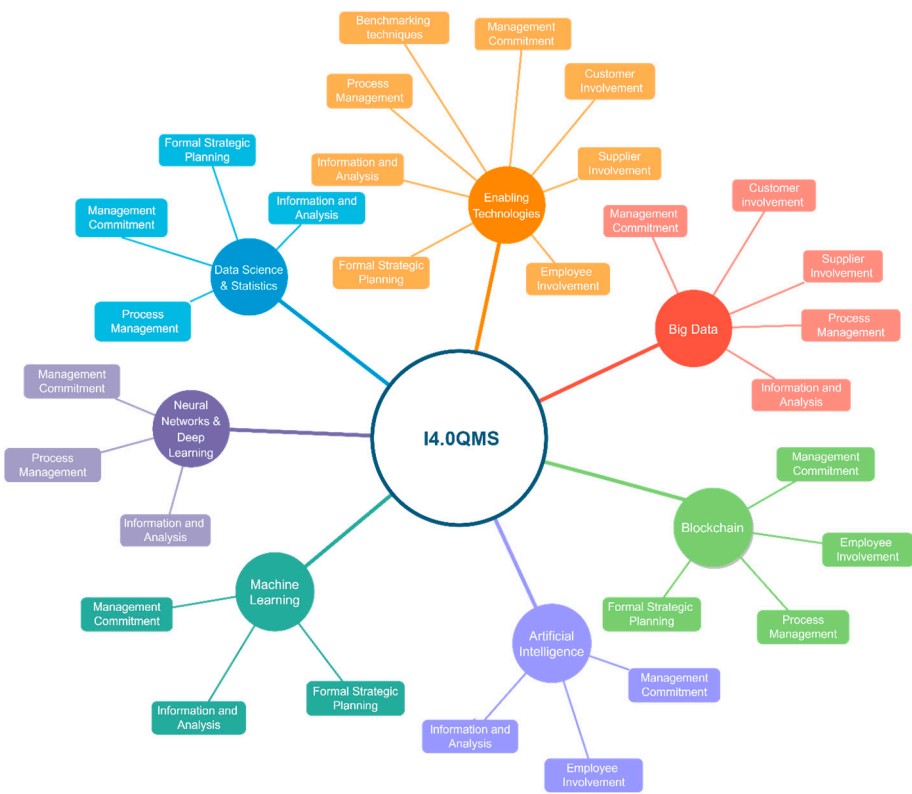

**Figure 2.** Relationship between Quality Management System practices and Industry 4.0 Tools, based on [13].

### 3.2. I4.0QMS—How to Achieve It?

Cognitive Engineering can have an influence on QMS practices. In the various phases of QMS implementation, numerous viewpoints and methodologies of cognitive engineering are utilized. A more complex perspective centered on collaborative or shared work, where both humans and machines take advantage of each other's strengths and work around each other's weaknesses when doing collaborative tasks, has been advocated by

the cognitive engineering community [18]. Implementing new procedures and techniques is only a small part of improving the QMS within an organization. In order to produce high-quality, defect-free products and services at a quality and cost suitable for the market, and with reliable delivery, a set of behaviors, attitudes, and processes that prevail throughout the stages of design, production, service, marketing, and administration have to be implemented. With the intention of raising service quality, usability, overall performance, and giving organizations a more all-encompassing experience, the complexity of organization environments and the associated technologies has increased, which can overlap the cognitive demand of workers. Designing computer-based systems with the goal of enhancing human performance through enhancing and/or amplifying human intellect is known as cognitive engineering [18].

Cognitive engineering, from a methodological standpoint includes the techniques described in Figure 3. These techniques can have an influence on the QMS practices identified previously. From helping Management to make effective decisions or to apply Man-Machine Interface (MMI) to Process Management achieving better inputs from, for example, quality inspection. All results obtained in the shop-floor and/or business areas are important inputs to fulfill the QMS, and these techniques can aid not only from a conceptual point of view of the Quality Management System, but also from a practical perspective, being implemented in the processes that will generate input to it.

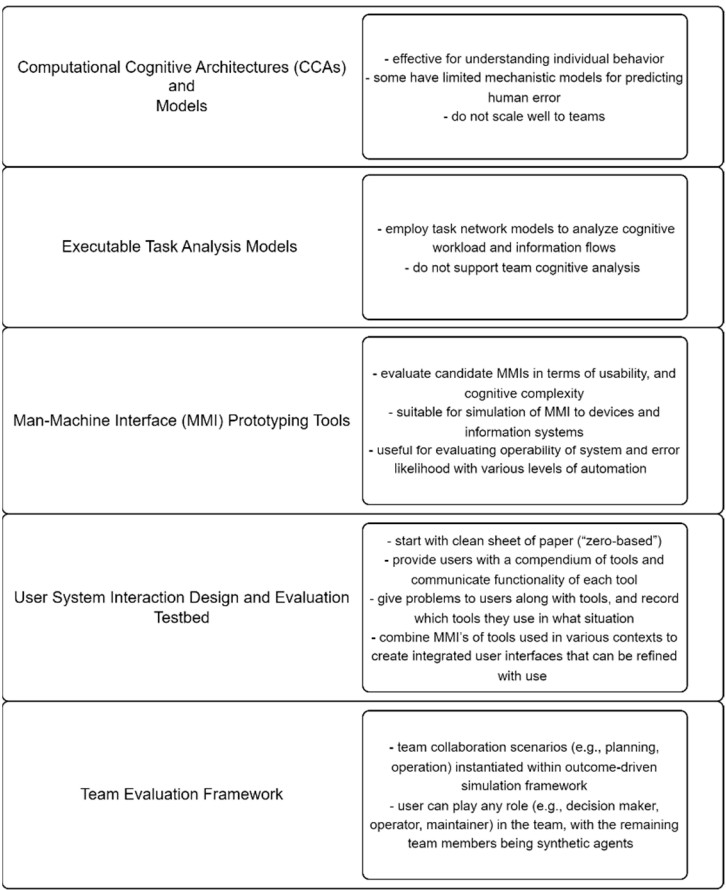

**Figure 3.** Cognitive Engineering Techniques based on [18,32].

Manufacturers and supply chains are further burdened by the level of standards needed for each component across hundreds of products. Additionally, the outsourcing of manufacturing facilities and the existence of numerous suppliers all over the world are to blame for the complexity of supply chains. It is difficult to build a standardized quality management system for the company when the contract manufacturers also have their own quality requirements.

Any quality problems would delay the launch of new products, which will result in a sizable revenue loss. These elements have made it more difficult for the manufacturing sectors to achieve deadlines for product development life cycles and deal with regulatory difficulties. These elements have given rise to the necessity of a more robust QMS, supported by the concept of Quality 4.0. Modern business practices have changed as a result of technological breakthroughs. The developed framework (Figure 4) blends the application of cutting-edge technologies with established QMS practices enhanced by CE in order to boost productivity, promote innovation, and achieve operational excellence.

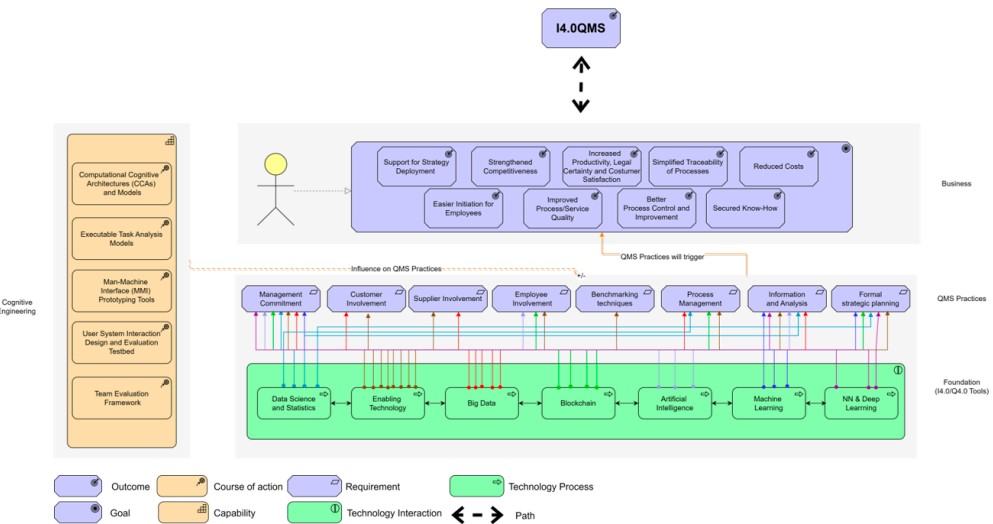

**Figure 4.** Conceptual Framework to aim in the application of a QMS 4.0—I4.0QMS.

Implementing Quality 4.0 initiatives may be hampered by a lack of digital quality expertise. Operators on the shop floor, who drive the quality efforts, must be familiar with a wide range of contemporary technology in order to execute Quality 4.0. At the firm level, a clear digital quality plan is required for implementation to be successful. There are more risks of execution failure the less clearly defined the plan is.

For this new framework, Figure 4, we consider the I4.0/Q4.0 tools as being our foundation, resulting from a technology interaction between the seven identified tools (Data Science and Statistics, Enabling Technology, Big Data, Blockchain, Artificial Intelligence, Machine Learning, Neural Networks & Deep Learning) These tools interact among themselves and can help in achieving one or more QMS practices, as explained in Figure 2.

Each one of these tools is a technology process connected with the previously identified QMS practices, which are influenced by the capabilities acquired from the CE techniques course of actions. These techniques are considered capabilities acquired by the workforce to take into action the QMS practices identified. Being influenced by these cognitive techniques, the professionals will be able to make more informed and clear decisions as their cognitive load is diminished and better strategies will be reached.

These QMS practices enhanced by the tools and the techniques will trigger the expected outcomes from a business point of view that constitute the goal. The following of this pathway will improve the chances to achieve, as an outcome, a better, more intuitive QMS—the I4.0QMS—making it possible to overview all the dispersed activities within the manufacturing environment gathered as one, and simplify the application of new technologies to the QMS activities.

## 4. Discussion

Many quality leaders are interested in implementing the Quality 4.0 idea in their organizations as a result of the excitement surrounding it. However, many of these leaders lack a clear understanding of how to implement Quality 4.0 methods into effect, and many projects never progress to the point of becoming an industry-ready, sustainable solution.

Another obstacle to implementation is the outcomes of quality data that are dispersed and segregated, this translates into some organizations having out-of-date QMS. The amount of data produced at different stages of the product lifecycle is enormous. The manufacturing data is dispersed throughout the company and is available in a variety of formats. Data quality challenges are brought up by such a large amount of data.

Four major issues facing the 4.0 era are covered in this article: Quality 4.0, Industry 4.0, Cognitive Engineering, and Quality Management System. To enable the effective deployment of an Industry 4.0-based QMS, these challenges must be properly understood and solved. The necessity of creating and implementing an I4.0QMS stems from the reality that, under the previous system, even minor adjustments required a time commitment that is nowadays inconceivable. A simple QMS would therefore inevitably result in utterly disorganized document storage in most expanding businesses. Such a method is not designed to effectively manage quality within a company. In a way that a conventional QMS is unable to, an I4.0QMS offers a broader perspective of all processes, services, and products delivered.

Since the activities are dispersed and segregated, it is difficult to develop a QMS that complies with the stringent performance, scalability, and availability requirements of industrial contexts. Adopting I4.0 technologies and integrating them with QMS practices could be difficult as a result. In this circumstance, the CE may be helpful by providing the workforce with the cognitive tools to make better decisions. An I4.0QMS will be more agile, flexible, and cost-effective.

By fusing the physical (QMS paper-based) and digital worlds, the I4.0QMS opens up new possibilities of productivity, growth, and information, giving businesses a thorough understanding of their most important assets and manufacturing procedures. What once required traditional tools and inefficient processes can now be performed by a number of I4.0 digital tools and the workforce can be supported by the CE techniques. This is a brand new approach that has not yet been taken into account by the literature, which focus more on the broad aspects of quality management [33,34].

An innovative conceptual framework is suggested in light of this investigation. The likelihood of success is increased by this thorough strategy. A clearly defined strategy guarantees that business imperatives are addressed, reduces adoption resistance, and makes it possible to choose projects wisely.

## 5. Conclusions

This manuscript aimed to analyze the relationship between quality management practices and Industry 4.0 technologies that improve quality are deeply studied and connected with cognitive engineering practices to develop an advanced framework.

A mixed inductive–deductive approach supports this research, which lead to the development of a conceptual framework that makes it easier to overview all the dispersed activities within the manufacturing environment gathered as one, and simplify the application of new technologies to the QMS activities, as it aimed. As limitations of this work the authors highlight the lack of recent publications on Cognitive Engineering field and Quality Management System.

This research lead to the development of a I4.0QMS framework, where the authors were able to achieve the three concepts that constitute the research objectives of this study.

1. The authors Target areas where QMS Practices will benefit from cognitive engineering techniques the most.
2. Focus on the core tools of Industry/Quality 4.0 most relevant to the QMS Practices.
3. Align Industry/Quality 4.0 objectives with QMS/Business Goals enhanced with the cognitive engineering basis.

Theoretically, Industry 4.0 will fundamentally alter the way that Quality Management Systems operate in the future years and will take center stage on the top management agenda, making the implementation of an I4.0QMS extremely helpful, as it will change the way organizations look at their practices and increase their competitiveness, enabling

them to implement better systems. From a practical point of view this I4.0QMS needs to be sharpened and future research should take this in consideration.

As future research, the authors highlight the importance of applying this conceptual framework from a practical perspective, which will aid in better understanding the influence between CE and the QMS practices, and also in the real application of the newly introduced I4.0QMS.

**Author Contributions:** Conceptualization, A.V.C.; methodology, A.V.C.; validation, A.V.C. and T.M.L.; formal analysis, T.M.L.; investigation, A.V.C.; resources, A.V.C. and T.M.L.; writing—original draft preparation, A.V.C.; writing—review and editing, A.V.C. and T.M.L.; visualization, T.M.L.; supervision, T.M.L. All authors have read and agreed to the published version of the manuscript.

**Funding:** This work has been supported by the project INDTECH 4.0 (POCI-01-0247-FEDER-026653), co-financed by the PT2020 and COMPETE2020 programs, and the European Union through the European Regional Development Fund (ERDF).

**Data Availability Statement:** Not applicable.

**Acknowledgments:** This work was supported in part by Fundação para a Ciência e Tecnologia (FCT) and C-MAST (Centre for Mechanical and Aerospace Science and Technologies), under project UIDB/00151/2020.

**Conflicts of Interest:** The authors declare no conflict of interest.

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
