# Peer review of "Quality 4.0 and Cognitive Engineering Applied to Quality Management Systems: A Framework"

_asi, doi:10.3390/asi5060115_

Round 1

Reviewer 1 Report

The article entitled Quality 4.0 and Cognitive Engineering Applied to Quality Management Systems: A Framework is correct in terms of content. The purpose of this paper is to develop a framework that aids in the application of a Quality 4.0 QMS.

However, personally, sometimes I was confused, and it wasn´t sufficiently clear for me some points in your work. Let me just give some suggestions in more detail, please.

The abstract must provide readers with a critical and more detailed view of the document, in addition to that quantitative results must be shown.

The keywords accurately reflect the content.

The paper is too short, in my opinion.

The authors can add a paragraph in the introduction to highlight the scientific contributions that authors propose through this paper. In the introduction, I would like to find the answers to some key questions, for example:

· Why your study is necessary? 

· What are the main literature gaps that you found that motivate you to plan this study?

· What is the main research originality/novelty?

· What are the main research goals?

The authors should reference more recent literature.

The research methodology should be better described.

Please consider adding a critical analysis of the figures.

In general, the discussions are limited, and little compared with other studies in international literature, this should be reviewed!

There are few citations and references in the paper in general, this is seen in the brief introduction that does not significantly cover the state of the art of the proposed topic.

In the conclusion section, the authors should enhance their practical and theoretical implications. 

The article presents interesting research results but needs to be improved before publication.

Author Response

Input from the reviewer:

The authors can add a paragraph in the introduction to highlight the scientific contributions that authors propose through this paper. In the introduction, I would like to find the answers to some key questions, for example:

  • Why your study is necessary? 
  • What are the main literature gaps that you found that motivate you to plan this study?
  • What is the main research originality/novelty?
  • What are the main research goals?

Authors response:

This remark was addressed a new chapter was added into the article addressing these points (line 221-248)

Input from the reviewer:

The authors should reference more recent literature.

Authors response:

The literature now includes more recent publications and the field where the authors noted a lack of recent publications is now better described (lines 250-294)

Input from the reviewer:

The research methodology should be better described.

Authors response:

The research methodology was better described (250-315)

Input from the reviewer:

Please consider adding a critical analysis of the figures.

Authors response:

The analysis of the figures was better described (317-336) (407-443)

Input from the reviewer:

In general, the discussions are limited, and little compared with other studies in international literature, this should be reviewed!

Authors response:

The discussion was reviewed and is now more extensive (456-494)

Input from the reviewer:

There are few citations and references in the paper in general, this is seen in the brief introduction that does not significantly cover the state of the art of the proposed topic.

Authors response:

The citations were reviewed and more references were added (541)

Input from the reviewer:

In the conclusion section, the authors should enhance their practical and theoretical implications. 

Authors response:

The conclusion now enhances the theoretical and the practical implications (495-526)

Reviewer 2 Report

Discussion section can be improved by comparing the study results with the existing literature. 

Author Response

Input from the reviewer:

Discussion section can be improved by comparing the study results with the existing literature. 

Authors response:

The discussion was improved and comparison with the literature included

Reviewer 3 Report

The topic undertaken by the authors is very interesting and important from many points of view. However, the article contains some errors and shortcomings that should be taken into account. 

Therefore, in order to improve the scientific quality of the article, some corrections should be made, as listed below:

1. The abstract should contain information on the results of the deliberations obtained by the authors.

2. Since the authors base their deliberations on QMS - they should also cite the relevant ISO standard.

3. A very poorly developed theoretical part with reference to a small number of references, which are very numerous in the analyzed area.

4. Vaguely formulated research goal and research problems - some can be found at the end of the Introduction section, some at various places in the Material and Methdos section.

5. In my opinion, the Materials and Methods section is written in an unclear manner, without correctly identifying and describing the research methods used.

6. The authors presented a rather complicated figure 2, but there is no explanation of it.

7. Figure 4 is hardly legible; there is also a lack of its broader description and commentary.

8. The Discussion section is too laconic. There are no references to the results of research by other authors.

9. The Conclusions section is incorrectly organized - first, the authors write down what method they used and what were the limitations, and then there is the purpose of the work (which is redundant at this point). There is no clear indication of what the authors have achieved.

10. As for an article presenting the framework of a problem, it is too short and supported by too few references.

11. References are prepared incorrectly - it should always start with the surname of the first author. In the case of this manuscript it is not uniform - sometimes the reference starts with the surname (e.g. No. 19), once with the initial of the first name (e.g. No. 1), and sometimes with the first name (e.g. No. 14

12. Sentences written by authors are often very long and complex, which makes them incomprehensible - please improve the style of the article.

13. Please re-examine the entire manuscript thoroughly and correct any grammatical, stylistic and linguistic errors.

Author Response

Input from the reviewer:

  1. The abstract should contain information on the results of the deliberations obtained by the authors.

Authors response:

Abstract was improved (line 10-23)

Input from the reviewer:

  1. Since the authors base their deliberations on QMS - they should also cite the relevant ISO standard.

Authors response:

Relevant ISO standard was referenced (line 541)

Input from the reviewer:

  1. A very poorly developed theoretical part with reference to a small number of references, which are very numerous in the analyzed area.

Authors response:

References were updated and an explanation of why the number of used references is smaller than the analyzed (line 286-303)

Input from the reviewer:

  1. Vaguely formulated research goal and research problems - some can be found at the end of the Introduction section, some at various places in the Material and Methdos section.

Authors response:

Research methodology and objectives were included (lines 249-315)

Input from the reviewer:

  1. In my opinion, the Materials and Methods section is written in an unclear manner, without correctly identifying and describing the research methods used.

Authors response:

Research methodology and objectives were included (lines 249-315)

Input from the reviewer:

  1. The authors presented a rather complicated figure 2, but there is no explanation of it.

Authors response:

Figure 2 was better explained (line 317-336)

  1. Figure 4 is hardly legible; there is also a lack of its broader description and commentary.

Authors response:

Figure 4 was better explained (line 407-443)

Input from the reviewer:

  1. The Discussion section is too laconic. There are no references to the results of research by other authors.

Authors response:

The discussion was improved and comparison with the literature included (456-477)

 Input from the reviewer:

  1. The Conclusions section is incorrectly organized - first, the authors write down what method they used and what were the limitations, and then there is the purpose of the work (which is redundant at this point). There is no clear indication of what the authors have achieved.

Authors response:

The conclusions were better organized and the achievement clearer (line 495-526)

Input from the reviewer:

  1. As for an article presenting the framework of a problem, it is too short and supported by too few references.

Authors response:

The article is more complete now, and went from having 10 pages to 17 pages

Input from the reviewer:

  1. References are prepared incorrectly - it should always start with the surname of the first author. In the case of this manuscript it is not uniform - sometimes the reference starts with the surname (e.g. No. 19), once with the initial of the first name (e.g. No. 1), and sometimes with the first name (e.g. No. 14

Authors response:

References were corrected accordingly (line 541)

Input from the reviewer:

  1. Sentences written by authors are often very long and complex, which makes them incomprehensible - please improve the style of the article.

Authors response:

Where possible this was addressed.

Authors response:

  1. Please re-examine the entire manuscript thoroughly and correct any grammatical, stylistic and linguistic errors.

Authors response:

Manuscript was re-examined in the best of the authors knowledge

Round 2

Reviewer 1 Report

The article entitled Quality 4.0 and Cognitive Engineering Applied to Quality Management Systems: A Framework” is correct in terms of content. The purpose of this paper is to develop a framework that aids in the application of a Quality 4.0 QMS.

However, personally, sometimes I was confused, and it wasn´t sufficiently clear for me some points in your work. Let me just give some suggestions in more detail, please.

The abstract must provide readers with a critical and more detailed view of the document, in addition to that quantitative results must be shown.

The keywords accurately reflect the content.

The paper is too short, in my opinion.

I suggest adding a paragraph in the introduction to highlight the scientific contributions that authors propose through this paper. Also, in the introduction, I would like to find the answers to some key questions, for example:

· Why your study is necessary? 

· What are the main literature gaps that you found that motivate you to plan this study?

· What is the main research originality/novelty?

· What are the main research goals?

The authors should reference more recent literature.

The research methodology should be better described.

Please consider adding a critical analysis of the figures.

In general, the discussions are limited, and little compared with other studies in international literature, this should be reviewed!

There are few citations and references in the paper in general, this is seen in the brief introduction that does not significantly cover the state of the art of the proposed topic.

In the conclusion section, the authors should enhance their practical and theoretical implications. 

The article presents interesting research results but needs to be improved before publication.

Reviewer 3 Report

After corrections the manuscript can be published in present form

Round 3

Reviewer 1 Report

I have no additional comments.